# Surface Deformation from Sentinel-1A InSAR: Relation to Seasonal Groundwater Extraction and Rainfall in Central Taiwan

Yi-Jie Yang [1], Cheinway Hwang [1,*], Wei-Chia Hung [1], Thomas Fuhrmann [2], Yi-An Chen [3] and Shiang-Hung Wei [1]

1   Department of Civil Engineering, National Chiao Tung University, 1001 Ta Hsueh Rd., Hsinchu City 30010, Taiwan; yangej@nctu.edu.tw (Y.-J.Y.); khung@itrige.com.tw (W.-C.H.); shianghung.cv97g@g2.nctu.edu.tw (S.-H.W.)
2   Positioning and Community Safety Division, Geoscience Australia, GPO Box 378, Canberra ACT 2601, Australia; thomas.fuhrmann@ga.gov.au
3   Department of Earth Sciences, National Central University, No. 300, Jhongda Rd., Jhongli City, Taoyuan County 32001, Taiwan; yianchen0822@g.ncu.edu.tw
*   Correspondence: cheinway@mail.nctu.edu.tw or cheinway@gmail.com

**Abstract:** Extracting groundwater for agricultural, aquacultural, and industrial use in central Taiwan has caused large-scale land subsidence that poses a threat to the operation of the Taiwan High Speed Railway near Yunlin County. We detected Yunlin subsidence using the Sentinel-1A Synthetic Aperture Radar (SAR) by the Small BAseline Subset (SBAS) method from April 2016 to April 2017. We calibrated the initial InSAR-derived displacement rates using GPS measurements and reduced the velocity difference between the two sensors from 15.0 to 8.5 mm/a. In Yunlin's severe subsidence regions, cumulative displacements from InSAR and GPS showed that the dry-season subsidence contributed 60%–74% of the annual subsidence. The InSAR-derived vertical velocities matched the velocities from leveling to better than 10 mm/a. In regions with few leveling measurements, InSAR increased the spatial resolution of the vertical velocity field and identified two previously unknown subsidence spots over an industrial zone and steel factory. Annual significant subsidence areas (subsidence rate > 30 mm/a) from leveling from 2011 to 2017 increased with the declining dry-season rainfalls, suggesting that the dry-season rainfall was the deciding factor for land subsidence. A severe drought in 2015 (an El Niño year) dramatically increased the significant subsidence area to 659 km². Both InSAR and leveling detected similarly significant subsidence areas in 2017, showing that InSAR was an effective technique for assessing whether a subsidence mitigation measure worked. The newly opened Hushan Reservoir can supply surface water during dry seasons and droughts to counter rain shortage and can thereby potentially reduce land subsidence caused by groundwater extraction.

**Keywords:** Hushan Reservoir; land subsidence; Sentinel-1; SBAS; Yunlin

## 1. Introduction

Land subsidence is the downward movement of the ground's surface caused by aquifer compactions, dewatering, and oxidation of organic soils, natural settlements of soils, and dissolution and collapse of earth materials [1]. In Taiwan, land subsidence is largely due to groundwater extraction. Recent climate change-induced irregularities in rainfall [2] and high demands for water use in the semiconductor, agricultural, and aquacultural industries have accelerated the use of groundwater [3], especially in Western coastal areas. Since the early 1990s, the Water Resources Agency (WRA) of Ministry of Economic Affairs (MOEA) of Taiwan have started to monitor and mitigate land subsidence

in Taiwan. Multiple techniques have been used for monitoring, including leveling, the use of global positioning system (GPS), multi-layer compaction monitoring well (MLCW), and interferometric synthetic aperture radar (InSAR) [4].

For long-term monitoring of subsidence, leveling and GPS are typically used. Leveling is accurate but time-consuming, and it can only provide time-lapsed displacement measurements in a network that has a limited number of benchmarks. GPS can measure three dimensional point displacements, but the precision of the vertical displacement from GPS is poor compared to that from leveling. A MLCW can detect compactions at different depths, which are important for understanding the mechanisms of land subsidence. As MLCWs are costly, only few are installed in Taiwan. These shortcomings in spatial and temporal resolutions and cost can be mitigated by InSAR, which has been widely used for monitoring land subsidence around the world. Many techniques have been used for the InSAR detection of land subsidence in Taiwan, e.g., Differential InSAR (DInSAR) [4,5], Persistent Scatterer Interferometry (PSI) [5–7], Temporarily Coherent Point SAR Interferometry (TCPInSAR) [4], and Small BAseline Subset (SBAS) [8].

Yunlin (see Figure 1 for Yunlin's location in Taiwan and a hydrogeological profile) is a coastal county in central Taiwan and is the focus area for this paper on land subsidence. Yunlin is situated over the Chuoshui River Alluvial Fan (CRAF) in central Taiwan that has long experienced significant land subsidence [6] (significant subsidence is defined as the case in which a subsidence rate is greater than 30 mm/a). The Taiwan High Speed Railway (THSR) passes through several subsidence-stricken townships of Yunlin, causing a major concern regarding the risk of THSR operation [9]. According to a recent land subsidence report [3], Yunlin's coastal townships, such as Mailiao Township, experienced severe land subsidence from 1992 to 1999; however, after 1999, the subsidence rates in coastal areas have been significantly reduced. In contrast, from 1999 to 2017, Yunlin's inland areas, such as Huwei, Tuku, Yuanchang, and Baozhong Townships, experienced increasingly severe land subsidence [3]. Additional risks of land subsidence, which are not only limited to Yunlin, and include flooding and change of water flows.

There have been many studies monitoring land subsidence in Yunlin. A study applying InSAR analysis on ERS and Envisat data shows that the mean land subsidence rate in Yunlin was about 30 mm/a over 1995–2001, rising to 50 mm/a from 2005 to 2008 [8]. Tung and Hu [10] showed that Tuku and Yuanchang Townships experienced the largest subsidence rates in Yunlin, reaching 78 mm/a along the line of sight (LOS) direction of ERS between 1996 and 1998. Using Envisat data, Hung et al. [6] showed that the largest land subsidence rate from 2006 to 2008 was about 70 mm/a in Huwei and Tuku Townships. Finally, Ge et al. [11] used ALOS/PALSAR to show that land subsidence rates in Huwei Township were over 100 mm/a between December 2006 and February 2011, the highest reported rates of subsidence in Yunlin.

Most InSAR studies in Yunlin focused on long-term and large-scale monitoring of subsidence. Only few studies in Yunlin show how InSAR can detect small-scale subsidence that have not previously been identified by leveling and GPS measurements [6,10,11]. In addition, most subsidence monitoring results in Yunlin only show long-term (over few years) subsidence behaviors without detecting the ups and downs of land surface in response to seasonal groundwater fluctuations [12], which can be used for groundwater management to mitigate land subsidence in dry and wet seasons. Some parts of Yunlin are such highly responsive areas that information is needed regarding their seasonal surface displacements, particularly because of the concern over the THSR's structural safety during droughts.

In April 2014, the European Space Agency (ESA) launched the Sentinel-1A SAR satellite mission. Sentinel-1A features a high revisiting frequency (12 days in the area of interest), high spatial resolution (approx. 5m × 20m along range and azimuth directions), wide swath (250 km), and free cost. In this study, we use SAR images from Sentinel-1A from April 2016 to April 2017, along with the SBAS approach, which is suit for monitoring the temporal evolution of surfaces in non-urban areas like Yunlin [13], to determine both the annual and seasonal displacements in Yunlin. We use data from GPS to calibrate the initial InSAR result to achieve a final InSAR velocity field, which was assessed by

GPS and leveling. The calibration by GPS removes systematic errors in the InSAR measurements due to orbit errors, long-wavelength atmospheric effects, and other sources. Our results cover the entire Yunlin County and show how land surface fluctuates with rainfall. We also show spots of subsidence not identified by leveling, and advise the responsible agency to install a local leveling network to monitor detailed land displacements in these areas. Finally, we show how rainfall in dry seasons affect areas of significant land subsidence, and how a new reservoir in Yunlin may be a measure of subsidence prevention by supplying surface water to counter rainfall shortage.

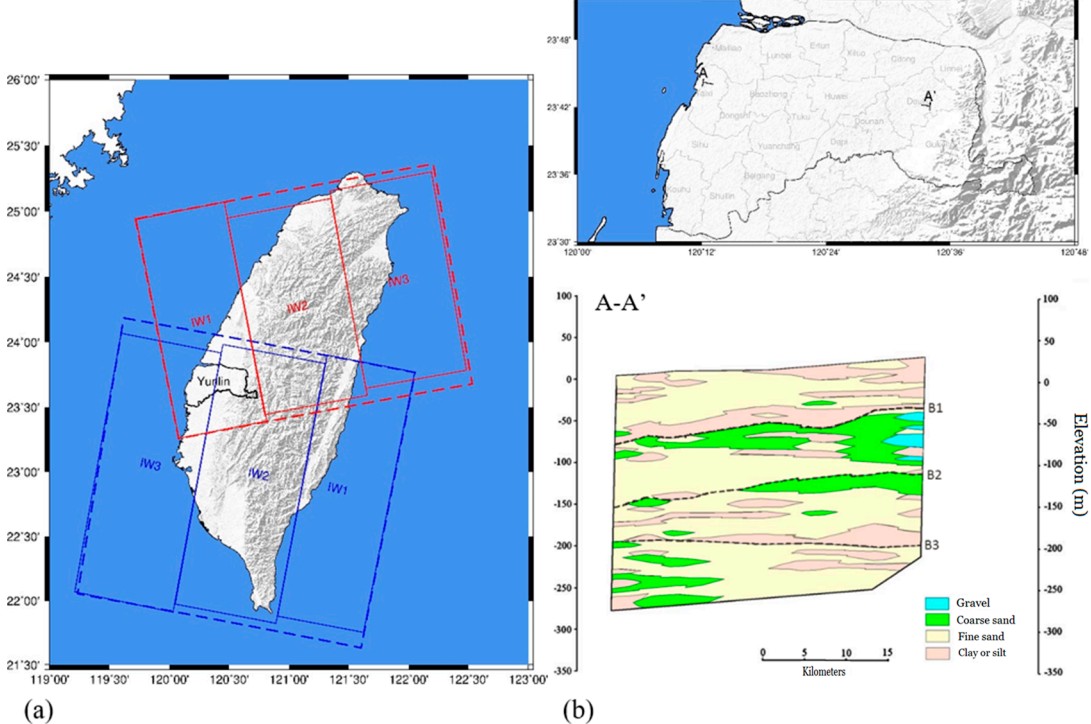

**Figure 1.** (**a**) The Sentinel-1 scene in the ascending and the descending tracks covering Yunlin. The red and blue dashed rectangular frames show the coverage of Frame 74 along ascending Track 69, and the coverage of Frame 510 along descending Track 105 in the IW mode, respectively. The solid rectangular frames show the three subswaths. (**b**) The distribution of four aquifers along a West–East profile (A-A') in the Choushui River Alluvial Fan (CRAF), modified from Reference [14]. Dashed lines B1–B3 mark the aquifer boundaries (Aquifer 1, 2, and 3 are above B1, between B1 and B2, and between B2 and B3, respectively; Aquifer 4 is below B3). The vertical axis for A-A' (bottom right) shows the aquifer elevations in m.

## 2. Data

### 2.1. SAR Dataset

Sentinel-1 is a C-band radar mission of the ESA in a two-satellite constellation, including Sentinel-1A and Sentinel-1B. Sentinel-1A was launched on April 2014 and Sentinel-1B was launched on April 2016. Both satellites are in the same orbital plane, but are separated by 180°. Thus, the revisiting intervals could be reduced from 12 to 6 days. Sentinel-1's orbit is sun-synchronous and has an altitude of 693 km and an inclination of 98.18°. From April 2016 to April 2017, Sentinel-1B did not provide SAR data covering Taiwan. This study uses 22 SAR images in the ascending track from Sentinel-1A from April 2016 to April 2017. The dates of the scenes are shown in Table 1. There are only 13 SAR images from the descending track over our study area (the coverage and the dates of the scenes are shown in Figure 1a and Table 1) and our processing results show that they produced poor qualities, and thus the images from the descending track were not used in this study (also see Section 3).

**Table 1.** The list of Sentinel-1A SAR images.

| Geometry | Scene No. | Date | Scene No. | Date |
|---|---|---|---|---|
| Ascending Track: 69 Frame: 74 | 1 | 04/14/2016 | 12 | 12/22/2016 |
| | 2 | 05/08/2016 | 13 | 01/03/2017 |
| | 3 | 06/01/2016 | 14 | 01/15/2017 |
| | 4 | 07/19/2016 | 15 | 01/27/2017 |
| | 5 | 08/12/2016 | 16 | 02/08/2017 |
| | 6 | 09/05/2016 | 17 | 02/20/2017 |
| | 7 | 09/29/2016 | 18 | 03/04/2017 |
| | 8 | 10/11/2016 | 19 | 03/16/2017 |
| | 9 | 10/23/2016 | 20 | 03/28/2017 |
| | 10 | 11/04/2016 | 21 | 04/09/2017 |
| | 11 | 11/28/2016 | 22 | 04/21/2017 |
| Descending Track: 105 Frame: 510 | 1 | 04/16/2016 | 8 | 12/12/2016 |
| | 2 | 06/27/2016 | 9 | 01/05/2017 |
| | 3 | 07/21/2016 | 10 | 01/29/2017 |
| | 4 | 08/14/2016 | 11 | 02/22/2017 |
| | 5 | 09/07/2016 | 12 | 04/11/2017 |
| | 6 | 10/01/2016 | 13 | 04/23/2017 |
| | 7 | 11/18/2016 | | |

Sentinel-1's mission provided data in different modes. In this study, we used the level-1 Single Look Complex (SLC) data from the Interferometric Wide swath (IW) mode, which had three separate subswaths. In Yunlin, we used the IW1 data from Track 69 and Frame 74, as shown in Figure 1a. The solid rectangular frames in Figure 1a show the three subswaths, IW1, IW2, and IW3 in the scene.

*2.2. GPS and Leveling Datasets*

The GPS data were used for calibrating the initial InSAR result from Sentinel-1A and to assess the final InSAR result. We used data from 14 continuous GPS stations in Yunlin (Figure 2), operated by WRA and the Central Weather Bureau (CWB) [3]. As an example, Figure 3 shows the time series of coordinate changes in the East, North, and vertical directions at station LNJS in Linnei Township (see Figure 3 for the location of LNJS, and Figure 2 for Linnei). The GPS time series in Figure 3 and those for InSAR calibration in this paper were the daily coordinates computed using Bernese 5.2 based on the IGS precise GPS orbits and observed earth rotation parameters. The Bernese outputs showed that the standard errors of the daily coordinates were about 3–5 mm. The orange lines show the best fitting line for the time series and represent the displacement rates from April 2016 to April 2017. The three-dimensional rates were at the level of several mm/a, suggesting that LNJS experienced few surface displacements.

The leveling data were used to assess the vertical velocities from Sentinel-1A. There were 406 leveling benchmarks (Figure 2) with measurements conducted in April 2016 and in April 2017, which were provided by WRA [3]. The leveling measurements were collected once a year with the precision specifications as follows. The project funder (WRA) specified the following error criteria: (1) The allowable double-run misclosure needed to be below $2.5\sqrt{k}$ mm, where $k$ is the distance between two neighboring benchmarks in km, and (2) the allowable loop misclosure needed to be below $3\sqrt{k}$ mm, where $k$ is the length of a loop. Our actual measurement results [3] showed that, in the 2017 leveling survey, the misclosures of all the 22 loops were smaller than the allowable loop misclosure ($3\sqrt{k}$ mm), and the average standard deviation of the heights determined by precision leveling was 3.3 mm [3]. The height accuracy in the 2016 leveling survey was similar to that in 2017.

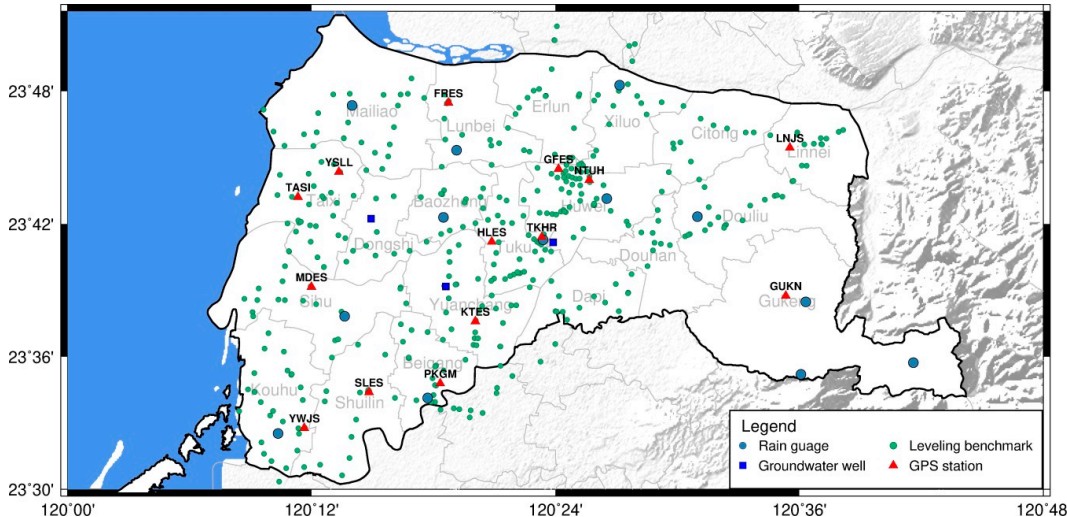

**Figure 2.** The 14 GPS stations (red triangles), 406 leveling benchmarks (green dots), 13 rain gauge stations, and 3 groundwater wells for analysis of the InSAR result in Yunlin. Names of Yunlin's townships are given in light gray in the background.

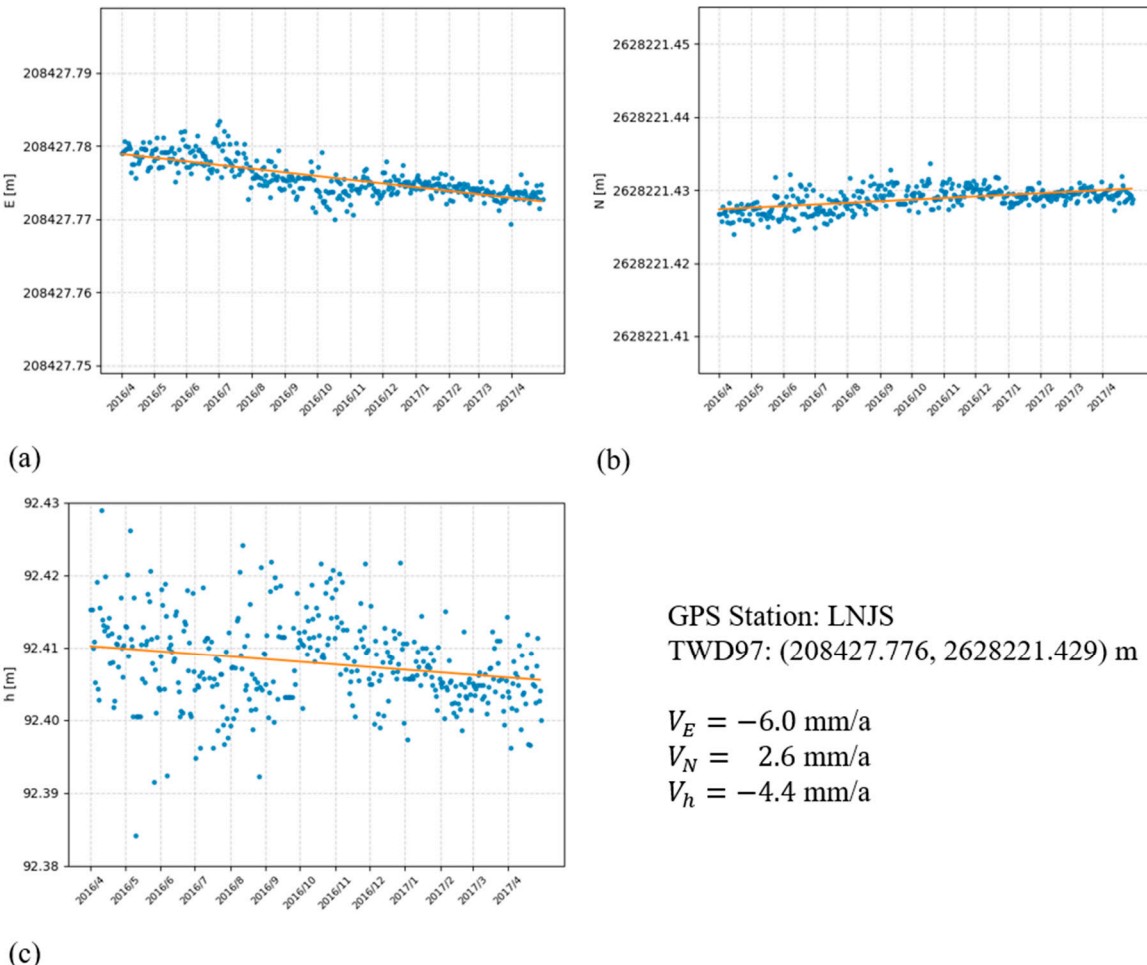

**Figure 3.** Time series of GPS-derived coordinate changes in the (**a**) East–West direction, (**b**) North–South direction, and (**c**) vertical direction at the LNJS station, which is a reference station for removing displacements offset in the initial InSAR result (see Section 4.1). The fitted lines (in orange) show the change rates ($V_E$, $V_N$, and $V_h$) from April 2016 to April 2017.

## 3. Method

In this study, we used measurements from InSAR to determine land displacement rates in Yunlin, which were then calibrated and assessed by the displacement rates from GPS and leveling. Figure 4 shows the workflow for computing and calibrating the InSAR displacement rates. First, the Sentinel-1A data were processed by an open source InSAR processing tool GMTSAR [15], which also provided a two-dimensional phase unwrapping tool SNAPHU [16]. We determined the LOS displacement rates from April 2016 to April 2017 and cumulative displacements in the wet and dry seasons by the SBAS method. SBAS used differential interferograms acquired at satellite positions with a small distance and a short-time interval, resulting in the so-called small baselines [17]. To reduce noises in the InSAR results, the spatial resolution of the final set of LOS displacement measurements was re-sampled to 10.5″ × 3″ along the longitude and latitude, which corresponded to a resolution of 297.5m × 85m in Yunlin. The re-sampling reduced the noise, but preserved the signals of land subsidence. Then, the GPS data (Figures 2 and 3) were used to determine velocities from April 2016 to April 2017 (the Sentinel-A time span). The three-dimensional velocity components ($V_N$, $V_E$, and $V_h$) from GPS were converted into the LOS velocity ($V_{LOS}$) using [18]:

$$V_{LOS} = \begin{bmatrix} \sin\theta\sin\alpha & -\sin\theta\cos\alpha & \cos\theta \end{bmatrix} \begin{bmatrix} V_N \\ V_E \\ V_h \end{bmatrix} \tag{1}$$

where $\theta$ and $\alpha$ are the Sentinel-1A incidence angle and the satellite heading angle (azimuth), respectively.

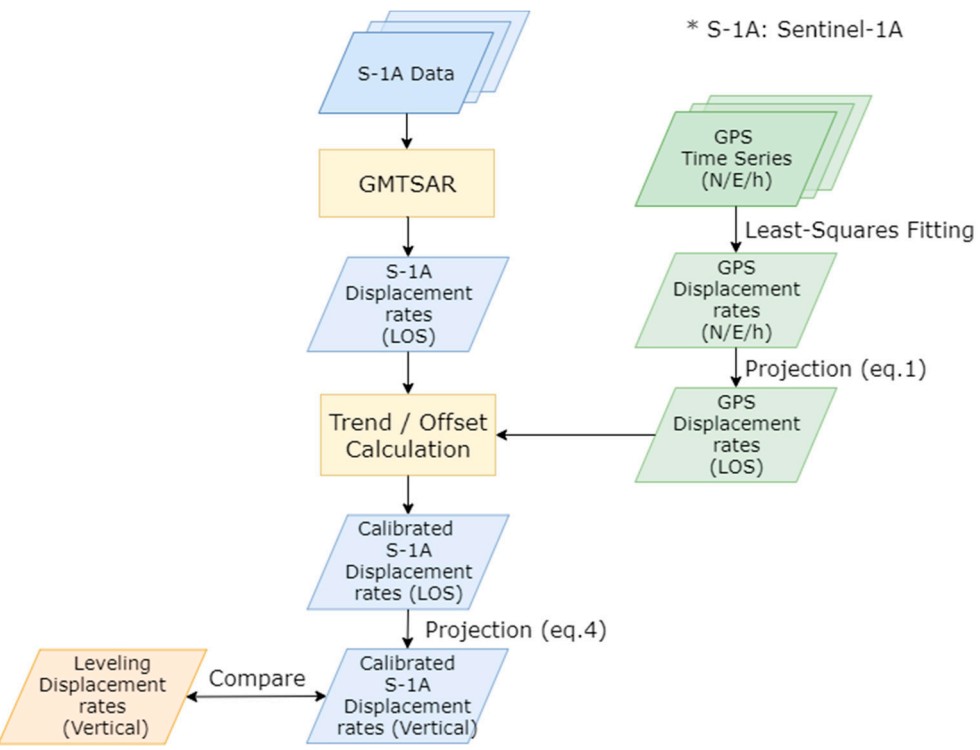

**Figure 4.** The workflow of the Sentinel-1A SAR determination of line of sight (LOS) and vertical velocities.

Next, all the GPS measurements were used to calibrate the initial InSAR result (see Section 4.1), which contained phase ramps in interferograms that were likely caused by orbit error and long-wavelength atmospheric effects [19,20]. The orbit error of InSAR could be decomposed into along-track (azimuth), across-track (range), and radial components [21]. The along-track orbit errors

could be regarded as timing errors, which are usually corrected during the coregistration of the two images in the InSAR processing. The across-track and the radial orbit errors can cause time-dependent baseline errors between the master and slave images and induce systematic phase ramps along the range and azimuth directions [18,21–24]. In addition, baseline errors can introduce errors in the across-track direction and cause phase ramps [21,25]. Such error-induced phase ramps can be modeled by a trend surface [22,24,26]; see Equation (2) below. A justification for using a trend surface was given by Hanssen [16], who showed that the ratio between the coefficients of a trend surface term and those of the quadratic surface was approximately $10^6$, suggesting that only the linear term was needed. In addition to orbit errors, the effect of the reference frame on InSAR results has been discussed [20]. As this effect results in a linear trend in the range direction, it is reduced when estimating a trend surface in the calibration step.

We used the GPS displacement rates along the LOS direction to estimate the required parameters in a trend surface that accounts for the systematic errors in the initial InSAR result. First, we calculated the mean LOS displacement rate from InSAR within a circular area around each GPS station and then calculated its difference with the GPS displacement rate (called the observed difference). We then set up the cost function related to the observed differences and modeled the differences [26,27]:

$$R^2 = \sum_{i=1}^{N} [\Delta d_i - (m_A + m_B x_i + m_C y_i)]^2 \tag{2}$$

where $\Delta d_i$ is the observed difference between GPS and InSAR at the GPS station, $i$, $x_i$, and $y_i$ are the East and North coordinates of the GPS station, $i$ and $N$ are the number of GPS stations, and $m_A$, $m_B$, and $m_C$ are the coefficients for the two-dimensional trend surface that absorbs the systematic errors in the initial InSAR result. Using the method of least-squares, we estimated $m_A$, $m_B$, and $m_C$ by minimizing $R^2$. Finally, we removed the linear trend from the initial InSAR displacement rates. After removing the trend, we used the GPS station LNJS as a reference station, and shifted all the InSAR displacement rates by the offset value between the InSAR velocity and the GPS velocity at LNJS to obtain the calibrated InSAR result (displacement rates) (see Section 4.1). As shown in Table 1, only a few images from the descending track were available and our tests showed that these images contained relatively large noises. In addition, our focus was to see whether a one-year Sentinel-1A result was sufficient to see the annual subsidence in the contemporary leveling results [6] and to experiment with detecting subsidence in wet and dry seasons. As such, we excluded the images from the descending track, and the InSAR result presented in this paper only used images from the ascending track (Table 1).

The InSAR result only contained displacement rates along the one-dimensional, LOS direction, which was not able to fully resolve the displacement rates in the three-dimension. While it is possible to resolve the two-horizontal components using SAR images from the ascending track and descending track, this was not carried out in this study due to the low number of images from the descending track and their noises (see the previous paragraph and Table 1), that could degrade our final InSAR result. Such LOS displacement rates have been converted into vertical displacement rates by neglecting horizontal velocities [12]. Fuhrmann and Garthwaite [28] have shown that the error in the vertical velocity, when converting LOS to vertical, depends on the incidence angle and the amount of horizontal motion. From a real-world local displacement phenomenon measured with Envisat data from multiple viewing geometries (four ascending and three descending passes), they found that an error of up to 67% of the maximum vertical motion was introduced into the projected vertical component when neglecting the horizontal component. However, the horizontal velocity field was almost of the same magnitude as the vertical in their case. In the Yunlin area, the vertical velocities and the horizontal velocities from April 2016 to April 2017 from GPS observations are shown in Figure 5. The vertical velocities were mostly over 20 mm/a and the horizontal velocities were mostly less than 5 mm/a. The

maximum error resulting from the projection of LOS into the vertical velocity induced by the horizontal velocity component was [28]:

$$\Delta V_e = \tan\theta(V_E \cos\alpha - V_N \sin\alpha) \tag{3}$$

which could be obtained by dividing the contributions of the $V_N$ and $V_E$ terms in Equation (1) by $\cos\theta$. Using an average incidence angle of 33.9° and a heading angle of −12.4°, we found that the average $\Delta V_e$ values at the GPS stations (Figure 5) were −2.4 mm/a, which was the average error due to neglecting the horizontal velocity components. As this error was relatively small, we neglected the horizontal velocities, i.e., $V_N = V_E = 0$, and converted the calibrated InSAR velocities to vertical velocities as (see Equation (1)):

$$V_h = V_{LOS}/\cos\theta \tag{4}$$

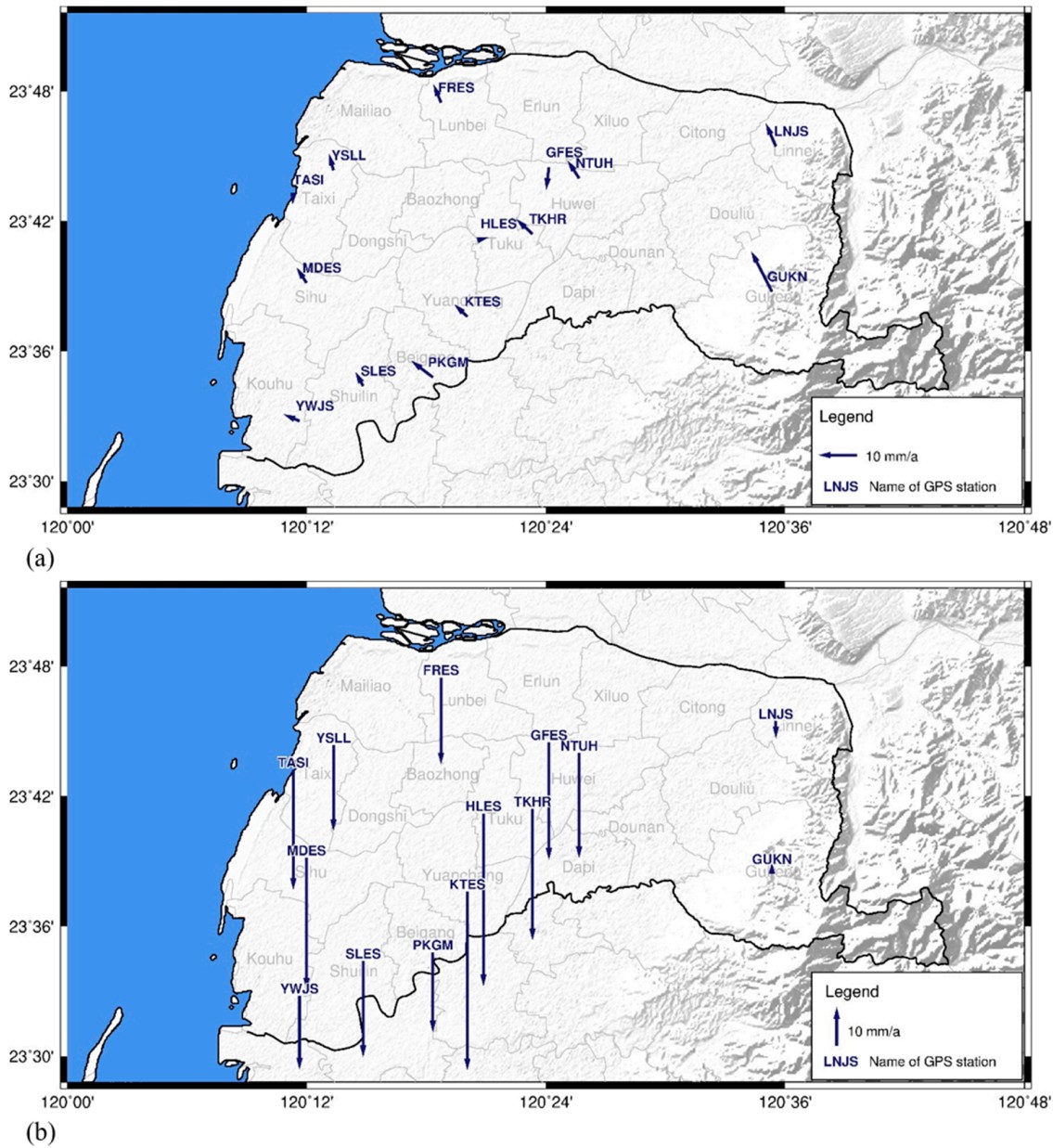

**Figure 5.** (**a**) The horizontal velocities and (**b**) the vertical velocities from April 2016 to April 2017 in Yunlin at the 14 GPS stations. The velocities are relative to the KMNMstation in Kinmen (at about 118.386°E and 24.464°N). The names of the townships in Yunlin are shown in the background.

This study compared the vertical velocities from InSAR with those from leveling to assess the calibrated InSAR displacement rates (see Section 4.3). Note that the workflow for the cumulative displacements in the wet and dry seasons (see Section 4.2) follows the same steps used for deriving the linear velocities from April 2016 to April 2017. The InSAR cumulative displacements were only assessed by GPS at selected stations, rather than the leveling measurements because leveling was only conducted once a year (in April; see Section 2).

## 4. Results

### 4.1. The GPS-calibrated InSAR Result from April 2016 to April 2017

First, we presented the removal of the systematic errors in the initial InSAR result using the GPS measurements. Figure 6 shows the differences between the initial InSAR velocities and the GPS velocities along the LOS directions at the 14 GPS stations. A positive value shows that the velocity derived from InSAR was smaller than that from GPS. The differences in Figure 6 indicate that the InSAR velocities are larger in the far range toward Eastern Yunlin, but smaller in the near range towards the Taiwan Strait, suggesting a linear trend of systematic errors in the initial InSAR velocities. In this case, the linear trend of InSAR velocities was 0.38 mm/a per km compared to the GPS velocities at the 14 GPS stations, and the offset value was –5.50 mm/a compared to the velocity at GPS station LNJS, which was located at a stable place with the smallest vertical and horizontal movements (mm/a level) in Yunlin.

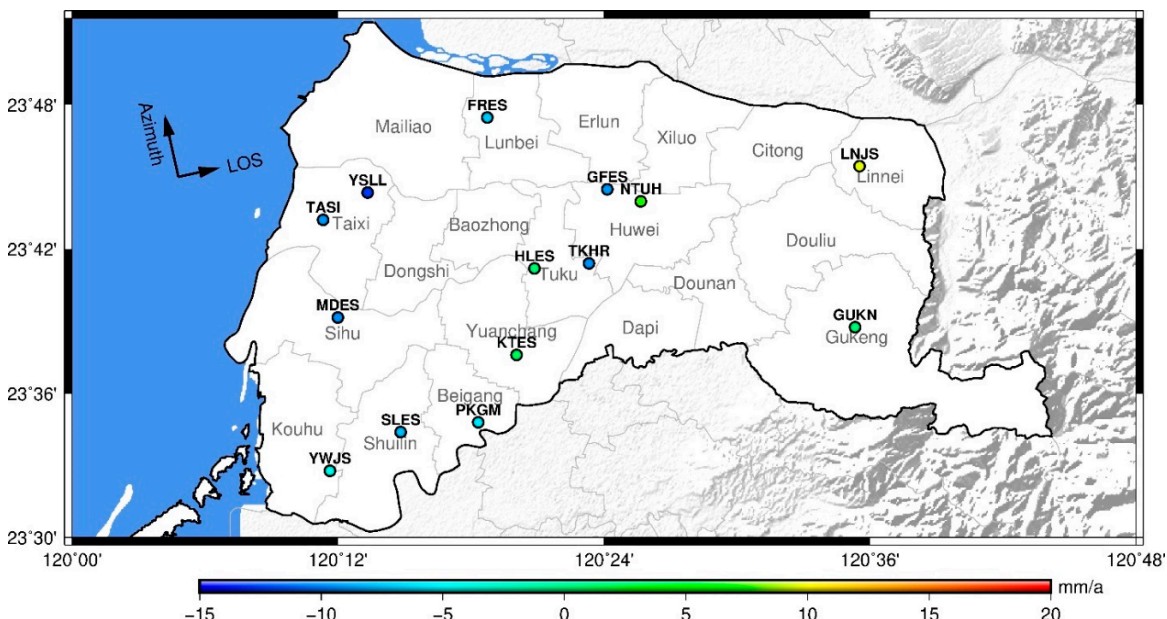

**Figure 6.** Differences between the initial InSAR and GPS velocities in the LOS directions, showing a tilting surface due to the systematic errors in the initial result.

Using the calibration described in Section 3, we obtained the final (calibrated) InSAR velocities (Figure 7). The RMS difference between the initial InSAR and GPS velocities was 15.0 mm/a and decreased to 8.5 mm/a after the GPS calibration. The differences between the GPS-calibrated InSAR velocities and the GPS velocities are mostly below 10 mm/a (Figure 7). In Figure 7, warm colors show land subsidence (negative LOS velocities) and cold colors show land uplift. The region with the largest land subsidence rate was in Southern-central Yunlin, near Tuku Township, where the subsidence rate reached 64.6 mm/a.

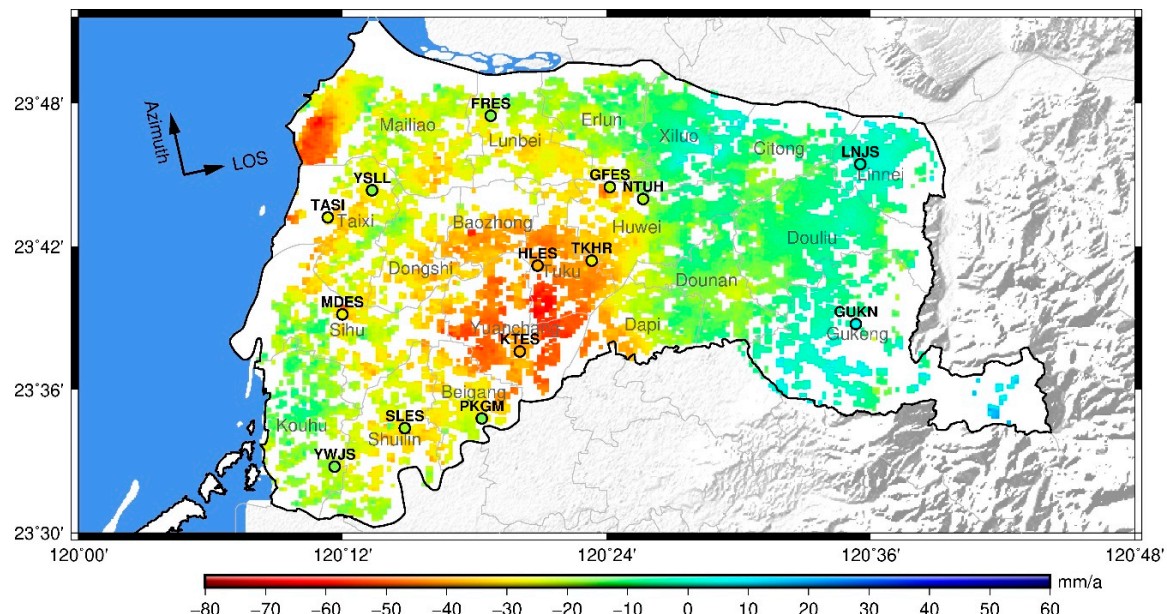

**Figure 7.** The GPS-calibrated InSAR velocities and GPS velocities (circles) in Yunlin from April 2016 to April 2017.

## 4.2. Displacements in the Wet and Dry Seasons

From April to September, the so-called wet season, Southwest monsoons and typhoons induce heavy rains throughout Taiwan. The rains provide abundant surface water and recharge groundwater, thus reducing the need for groundwater pumping and providing stability to the groundwater level. In contrast, rainfalls are rare from October to April in Taiwan (the dry season), leading to more groundwater pumping and larger land subsidence compared to the wet season. As an example, Figure 8 shows the monthly rainfalls from April 2016 to April 2017 in Yunlin, indicating a distinct change in the rainfall pattern between September 2016 and October 2016. In this study, we detected the different extents of land subsidence in the wet and dry seasons using the SAR interferograms (Figure 9). Each line in Figure 9 indicates one interferogram formed by two images in the wet or the dry season. There were 8 and 15 scenes in the two seasons, resulting in 16 interferograms for the wet season and 59 interferograms for the dry season. Each interferogram was chosen so that the resulting perpendicular baseline was less than 130 m and the two images were separated by a maximum time interval of 84 days.

Figure 10 shows the cumulative displacements from InSAR and the GPS stations in the wet and dry seasons, respectively. Figure 10a shows small land displacements (at few mm) in the wet season in most parts of Yunlin. The Eastern Yunlin is covered by hills and high mountains with dense vegetation. As such, the result from InSAR here was not reliable and was excluded in our analysis. Figure 10b shows distinct cumulative subsidence ranging from 30 to 50 mm in Southern-central and Southwestern Yunlin in the dry season. The small and large cumulative displacements in the wet and dry seasons were largely related to the ups and downs of groundwater levels in Yunlin [4] (see also the analysis in Section 5.1).

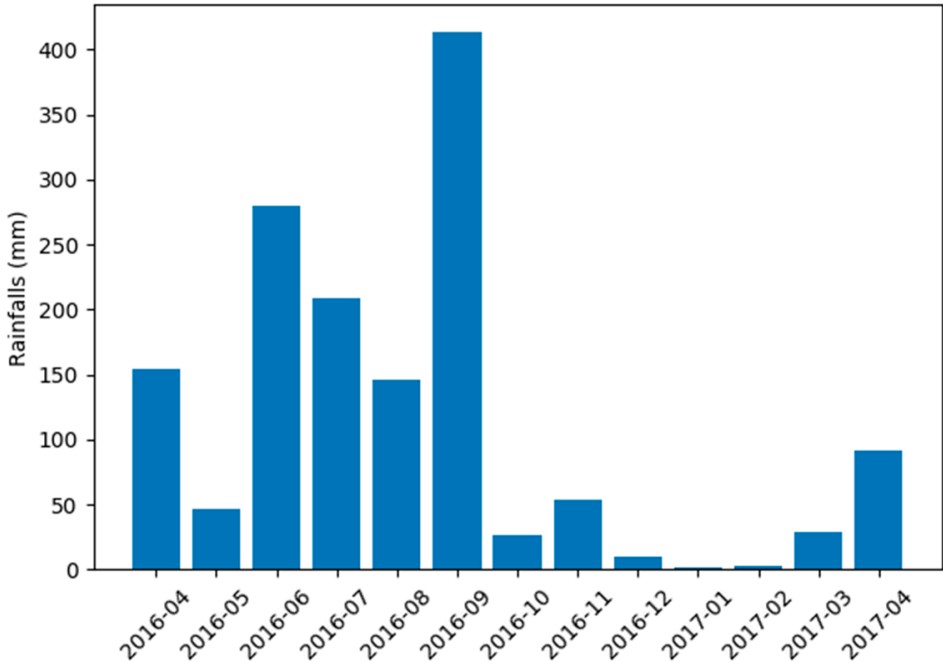

**Figure 8.** Monthly rainfalls in Yunlin, based on the data at 13 rain gauge stations in Yunlin provided by the Central Weather Bureau of Taiwan. To get such data, visit http://e-service.cwb.gov.tw/HistoryDataQuery/index.jsp [29].

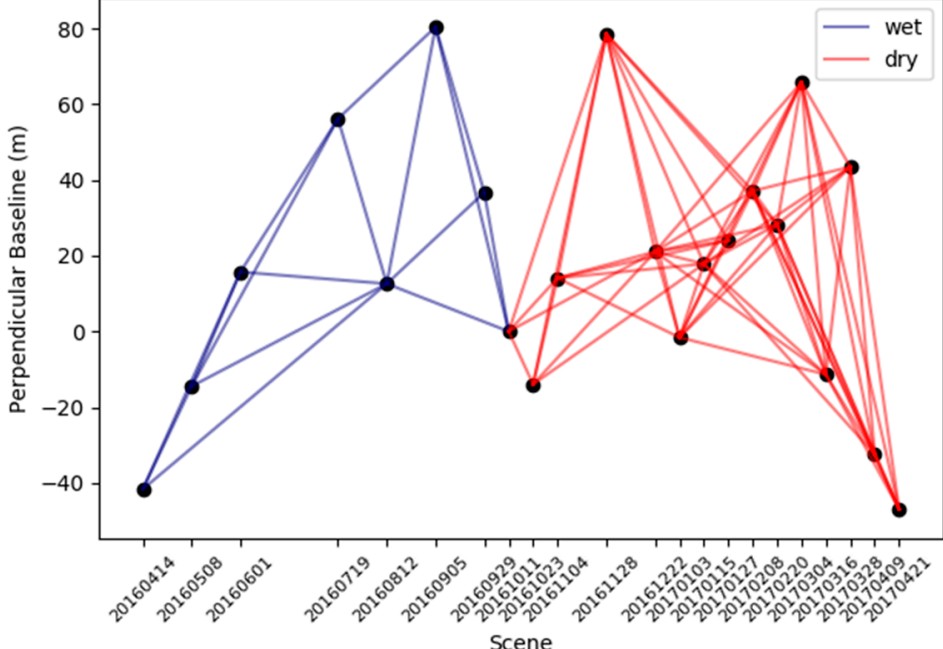

**Figure 9.** The paired SAR images forming the interferograms for determining displacements in the wet and dry seasons.

### 4.3. Assessing the Calibrated InSAR Result by Leveling

The leveling data form an independent dataset that can assess the GPS-calibrated InSAR result. Figure 11a shows the differences between the vertical velocities from InSAR and from leveling evaluated at the benchmarks in Figure 2. The InSAR velocities were consistent with those from leveling in most parts of Yunlin, except for Southwestern Yunlin. Figure 11b shows the scatter plot for the velocities from InSAR and leveling at the benchmarks. If the two sets of velocities were consistent

at the benchmarks, they would be on a line with a gradient of one (the gray line) in Figure 11b. In reality, inconsistencies existed between the two sets of velocities (the cyan line shows the linear relationship from the least-squares fitting to the two velocity sets). The RMS difference between the un-calibrated InSAR velocities and leveling velocities was 16.0 mm/a and reduced to 9.4 mm/a after the GPS calibration.

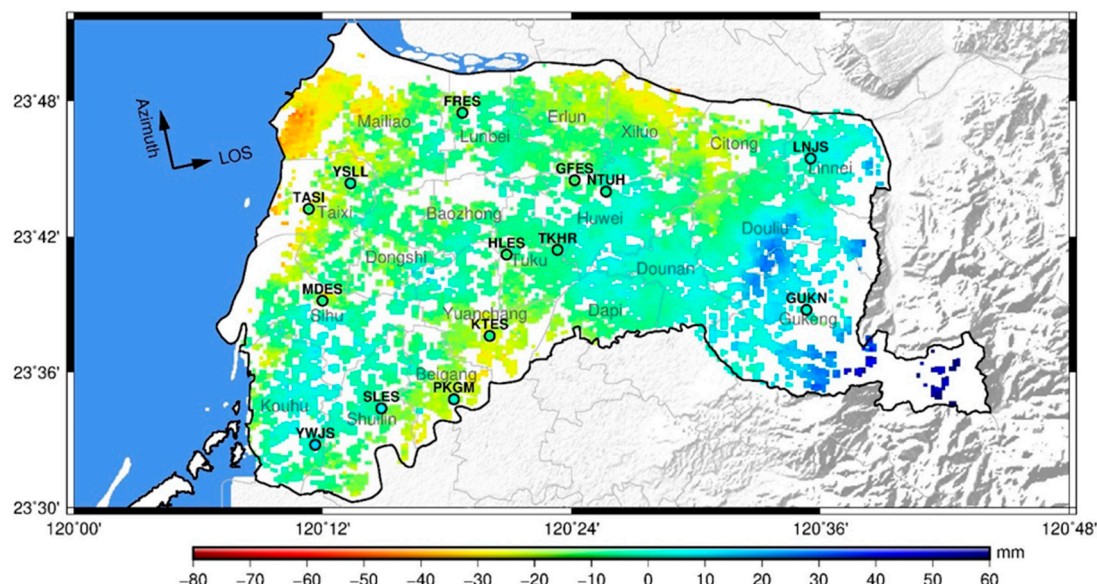

(a)

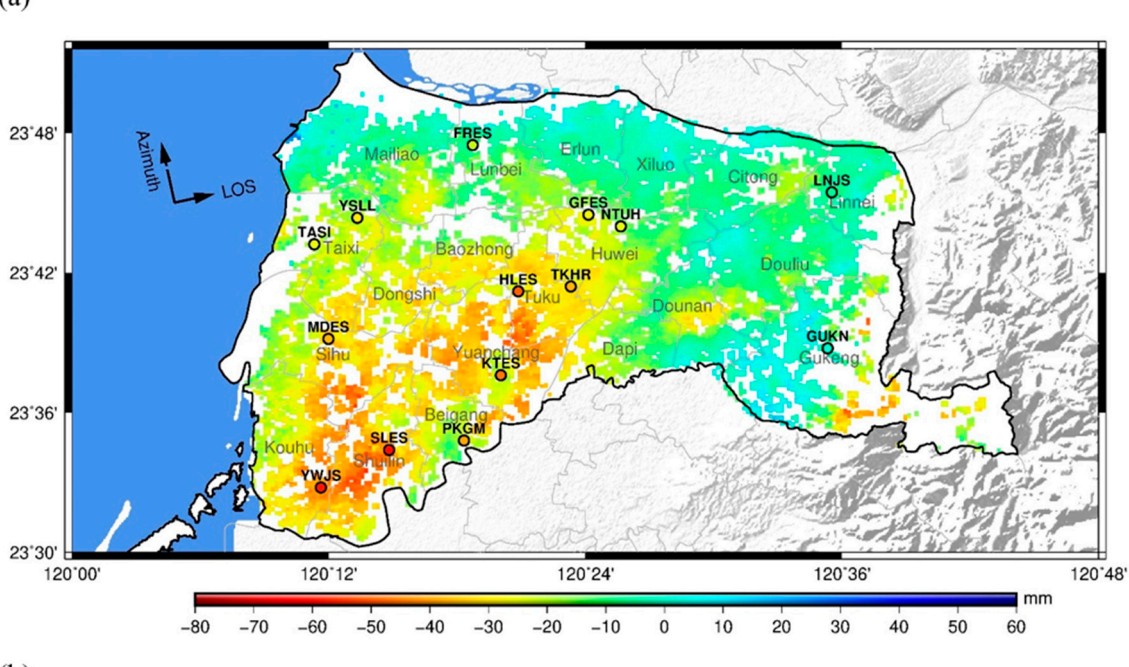

(b)

**Figure 10.** LOS displacements from the calibrated InSAR result and GPS measurements (point values with station names) in the (**a**) wet season (from April 2016 to October 2016) and (**b**) dry season (from October 2016 to April 2017). The anomalously large displacements in Figure 10a in Eastern Yunlin could be due to noises in the SAR images that are not removed in our data processing.

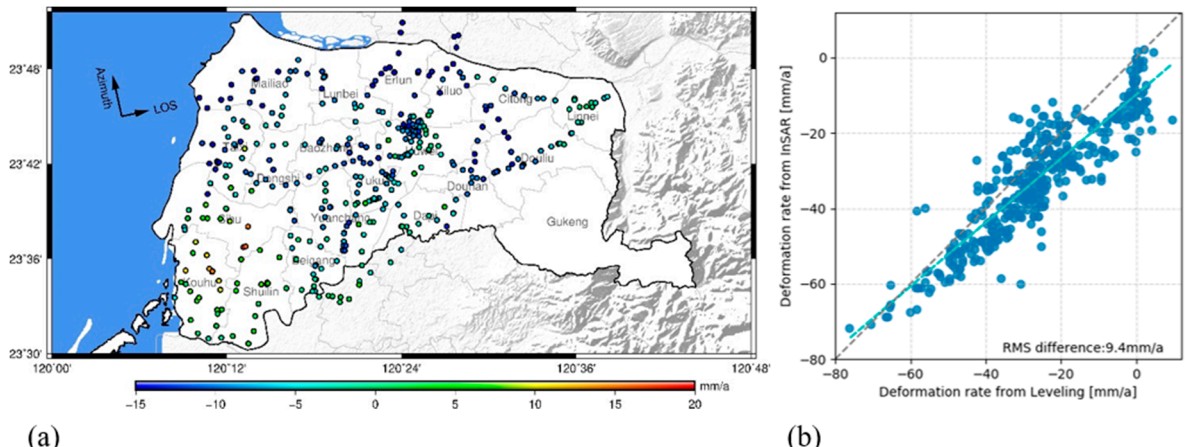

**Figure 11.** (**a**) Differences between the vertical velocities from leveling and from InSAR at leveling benchmarks. (**b**) The scatter plot for velocities from leveling and InSAR. The gray line is the "ideal" line when the InSAR and leveling results are the same, and the cyan line shows the actual linear relationship from the least-squares fitting to the two velocity sets.

## 5. Discussion

### 5.1. Land Subsidence in the Wet and Dry Seasons

The first discussion is about the InSAR-detected subsidence in the wet and dry seasons and its implication. First, we assessed the InSAR displacements in the two seasons by GPS measurements. Figure 12a shows the locations of five evenly distributed GPS stations for this assessment. These stations were located in Tuku, Yuanchang, Huwei, Baozhong, and Mailiao Townships, where the subsidence rates were relatively large. Figure 12b–f show the time series from InSAR and GPS. The overall patterns of the InSAR and GPS time series were similar, but some of the InSAR displacements were anomalous. At GPS stations FRES, GFES, MDES, and KTES (station names and townships are given in Figure 2), the time series showed sudden shifts with respect to the GPS time series on October 23, 2016 (roughly the transition time from the wet season to the dry season). However, in both the dry and wet season, the trends of subsidence from InSAR and GPS were similar. To show the consistency in the subsidence trend between the GPS time series and InSAR time series in different seasons, we shifted the time series of InSAR in the dry season in Figure 12b–f (red lines). The shifts in the InSAR time series in the dry season may have been caused by un-modeled, wet tropospheric delays induced by rain cells or by sudden, large increases of soil moisture that delayed radar signal propagation [30], unwrapping error [31], or due to other unknown error sources in our InSAR data processing.

Figure 13a shows the cumulative areal land subsidence values derived from InSAR (starting from April 2016) in the five townships and the rainfalls (Figure 8) at about one-month intervals in Yunlin. Figure 13b shows time series of groundwater level changes in Yunlin in the five townships. In general, the lowest groundwater tables occurred in April, while the highest tables occurred in September. The range of groundwater level changes in Figure 13b varied from one location to another. The slight land rebounds in the wet season were mostly due to rising groundwater that was recharged by rain. The dry season began in October with gradually decreasing rainfalls that resulted in increased land subsidence. Table 2 shows the contributions of the land subsidence in the two seasons in the five townships. The ratios of contribution of land subsidence in different seasons are calculated as:

$$Ratio_{wet} = \frac{Deformation_{wet}}{Deformation_{wet} + Deformation_{dry}}$$

$$Ratio_{dry} = \frac{Deformation_{dry}}{Deformation_{wet} + Deformation_{dry}}$$

(5)

where $Ratio_{wet}$ and $Ratio_{dry}$ are percentages of contributions of land subsidence in the wet and the dry seasons, respectively, and $Deformation_{wet}$ and $Deformation_{dry}$ are cumulative displacements in the two seasons.

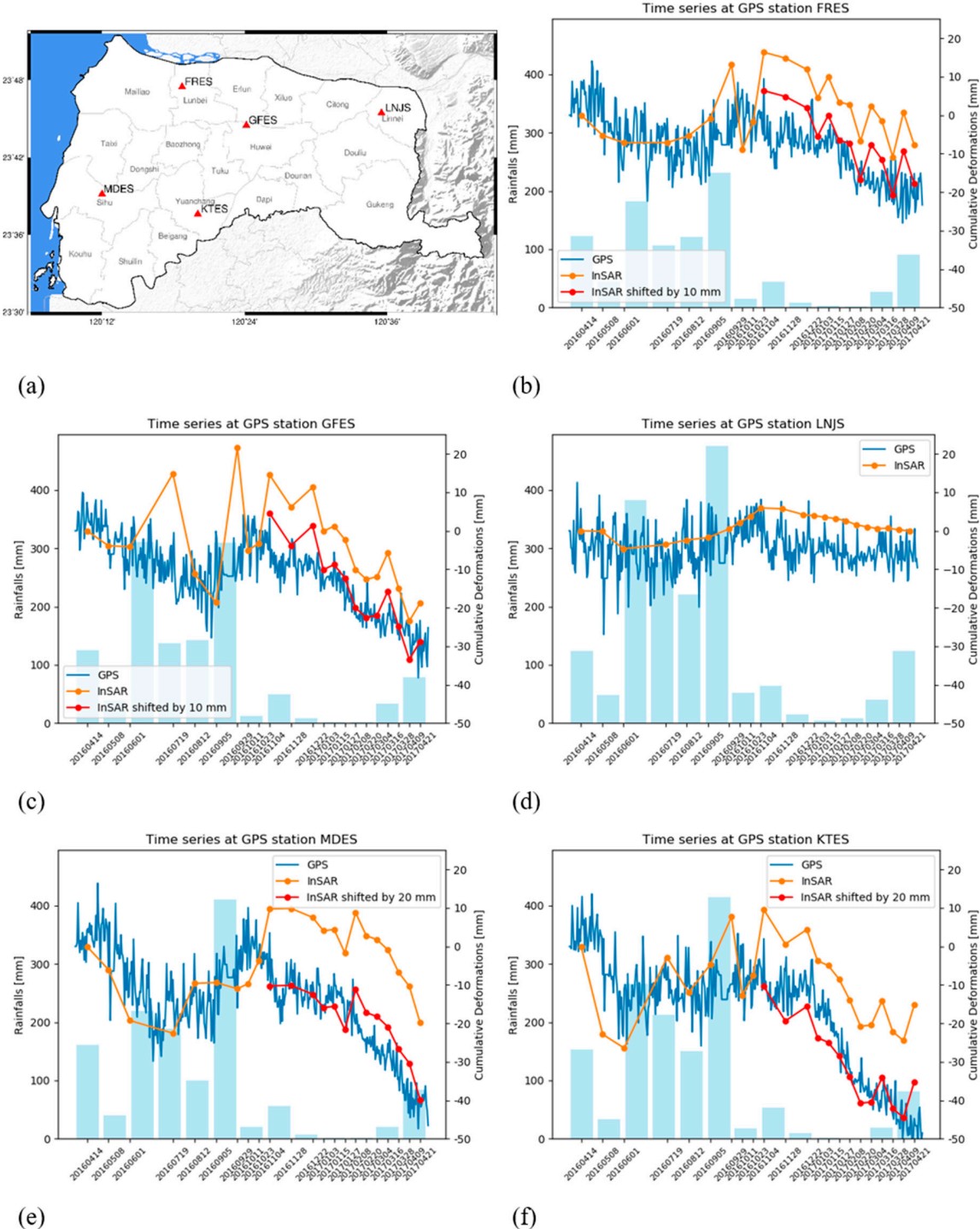

(a)

(b)

(c)

(d)

(e)

(f)

**Figure 12.** (**a**) The locations of five GPS stations where the time series of LOS displacements from GPS and InSAR are plotted to show subsidence contributions from the wet and dry seasons, (**b**–**f**) LOS displacements from GPS (blue) and InSAR (orange) at FRES, GFES. LNJS, MDES, and KTES, overlapped with monthly rainfalls averaged from all rain gauge stations in Yunlin (Figure 8). The shifted time series (red lines, with shifted subsidence values) are given to show the consistency between the LOS displacements from InSAR and GPS.

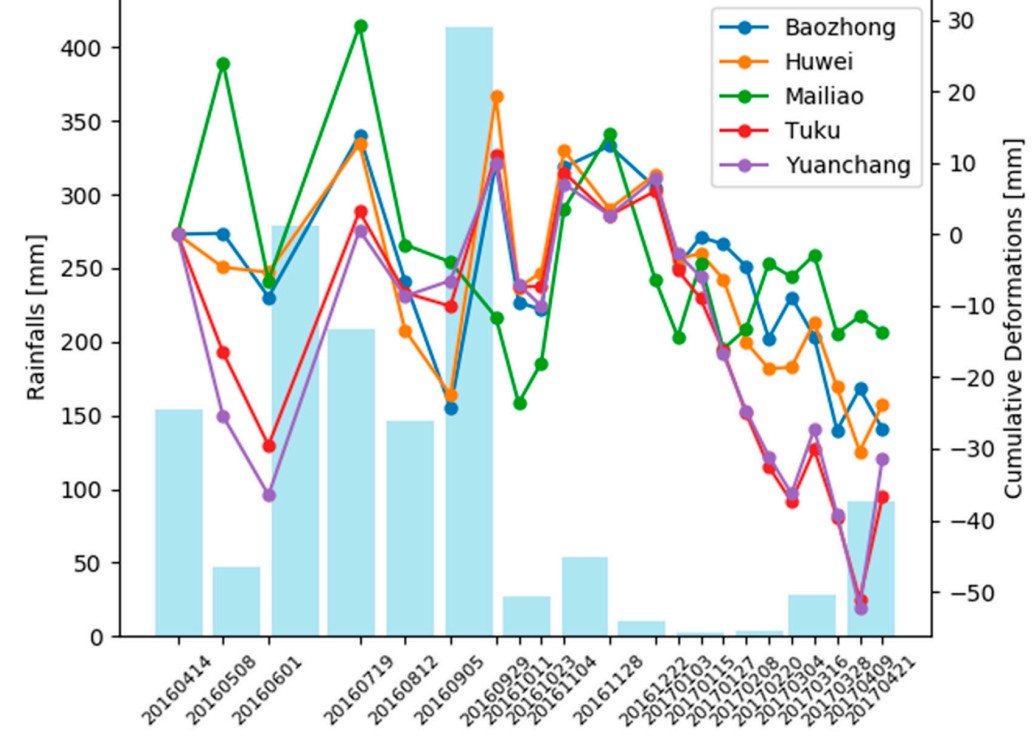

(a)

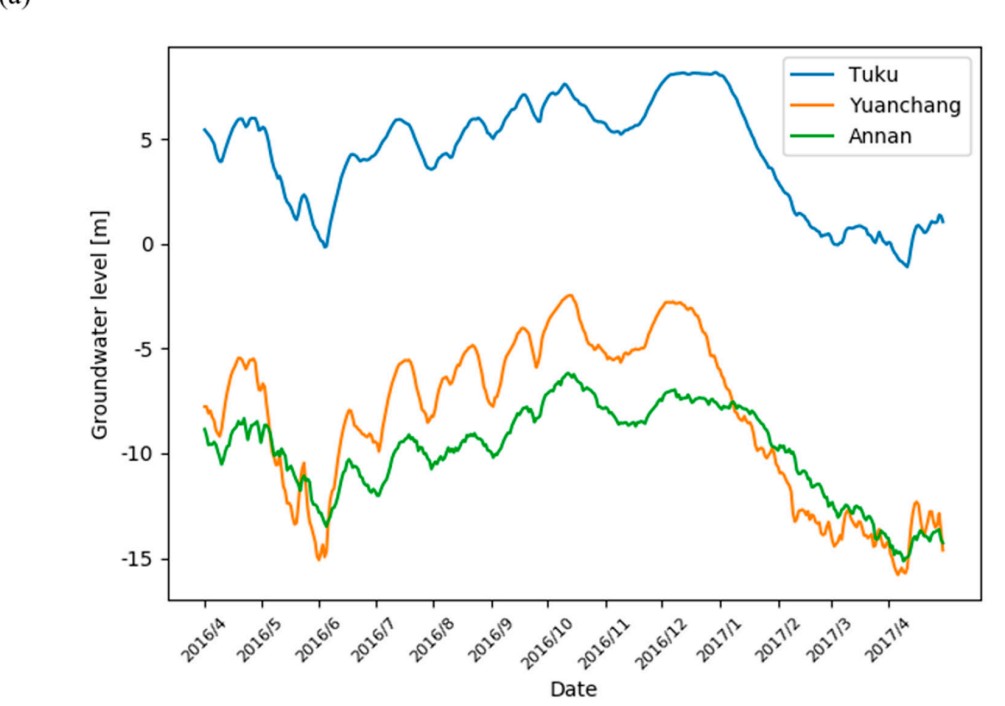

(b)

**Figure 13.** (**a**) Cumulative subsidence and monthly rainfalls in the most severe land subsidence locations of the five townships (see Figure 5), overlapped with the cumulative averaged rainfalls [29]. (**b**) Groundwater level changes at the three selected groundwater wells (see Figure 2, squares).

**Table 2.** The contributions of land subsidence in the wet and dry seasons in the most severe land subsidence locations of the five townships.

| Township | Wet Season | | Dry Season | |
|---|---|---|---|---|
| | Cumulative Displacements (mm) | Ratio (%) | Cumulative Displacements (mm) | Ratio (%) |
| Yuanchang | −16.97 | 27.52 | −44.70 | 72.48 |
| Tuku | −17.68 | 26.25 | −49.67 | 73.75 |
| Huwei | −18.44 | 37.22 | −31.11 | 62.78 |
| Baozhong | −18.12 | 39.23 | −28.07 | 60.77 |
| Mailiao | −44.26 | 80.94 | −10.42 | 19.06 |

It is believed that more groundwater extraction in the dry season caused a greater contribution of land subsidence in the five townships. The dry season contributed between 60% and 74% of the total subsidence (except Mailiao Township). The dry season contribution tended to increase with subsidence. For example, in Tuku, the cumulative subsidence was 49.67 mm in the dry season with a 73.75% contribution, compared to the cumulative subsidence of 28.07 mm in Baozhong that contributed 60.77% to the total subsidence. The percentages in Table 2 are consistent with a general impression that rainfall can slow down land subsidence due to increased surface water use and less groundwater extraction. One exception is Mailiao Township, where land subsidence in the wet season was larger than that in the dry season. Mailiao Township is a land reclamation area and it houses a large industrial zone. More data are needed to explain the different cumulative subsidence pattern in Mailiao.

Except Mailiao Township, Yunlin consists of largely agricultural and aquacultural areas. This study showed that InSAR can detect short-term (monthly) variation of land subsidence, which also showed that different industries in Yunlin use groundwater in different ways. We recommend InSAR be used to monitor land subsidence in areas with different industries. Such InSAR results can be used along with monthly measurements from MLCWs [4] to understand the mechanism of land subsidence. This joint use of remote sensing and subsurface data can help to initiate effective rules to govern the use of groundwater and surface water to mitigate land subsidence.

*5.2. Newly Identified Subsidence Spots in Yunlin by InSAR*

The next subject of discussion is the ability of InSAR to identify spots of subsidence not detected by leveling, thanks to InSAR's much higher sampling density compared to leveling (Figure 2). Here, we focused this ability of InSAR on the places that probably experience severe land subsidence, but have only few leveling benchmarks. The red rectangular box in Figure 14 shows one of the townships with the most severe land subsidence is Baozhong Township, and the zoom-in map is shown in Figure 15. The village of Shinhu in Baozhong Township shows a spot of potential large land subsidence. Note that the largest subsidence rate around Baozhong was 67.6 mm/a. We conducted an in-situ investigation in Shinhu Village and conclude that, although Shinhu is largely a residential and farming area, it houses a steel factory with a location that is consistent with the location of large land subsidence seen in Figure 15 (within the dashed circle). Thus, we suspected that the factory was the source of the land subsidence. It was suspected that the loading of steel and unauthorized groundwater pumping inside the factory contributed to soil compaction around the factory. Thus, it is recommended that a densified network of leveling benchmarks be deployed here to detect land subsidence for further actions.

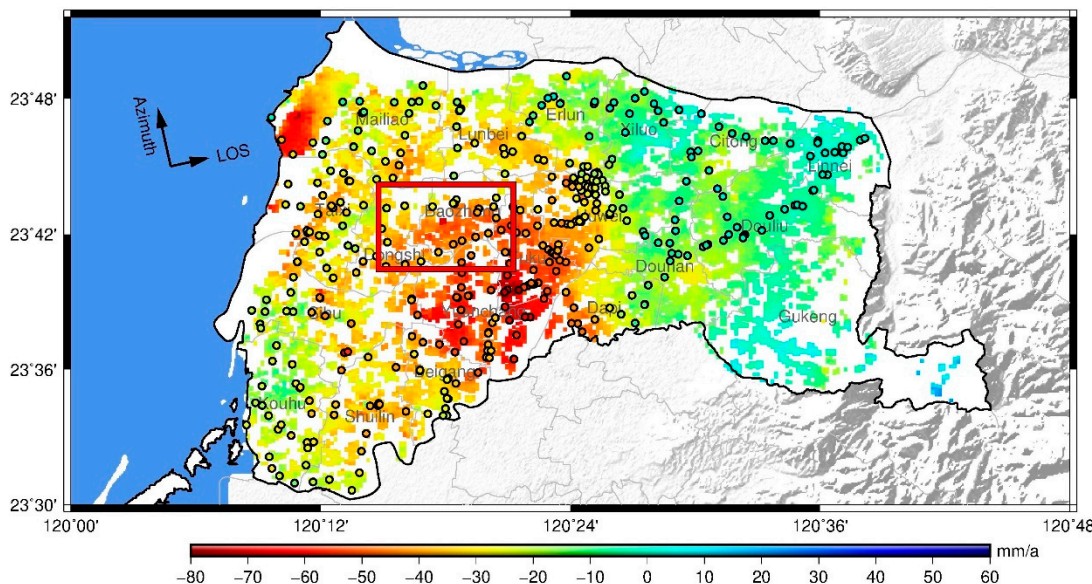

**Figure 14.** The vertical displacement rates from InSAR and leveling. The red rectangular frame shows the location of our in-situ investigation (see Figure 15). Note the large subsidence rates in Mailiao (dashed circle).

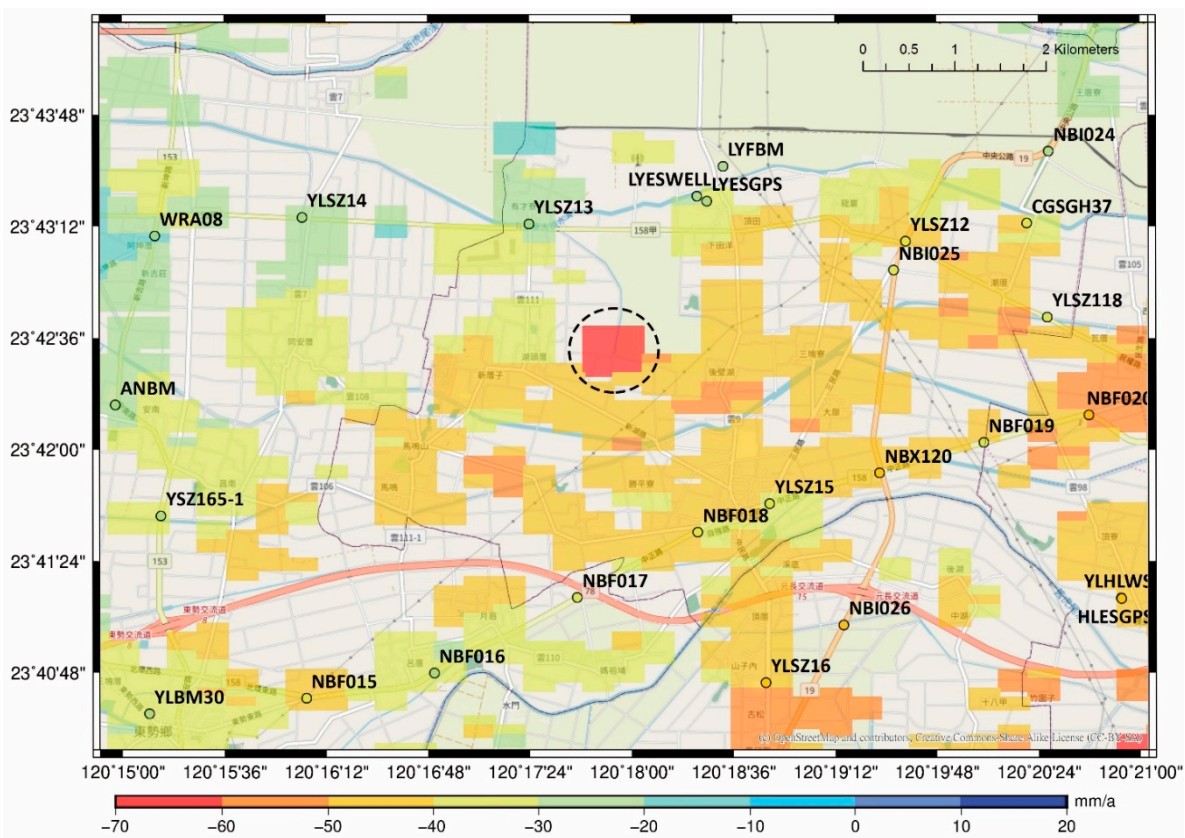

**Figure 15.** Vertical displacement rates from InSAR and leveling in the red rectangular box of Figure 14. Our field investigations indicate a steel factory at the center of the dashed circle, where the displacement rate is the largest.

Another spot with large InSAR-detected subsidence rates was the industrial zone in Mailiao Township (dashed circle in Figure 14). As shown in Figure 14, there are only seven leveling benchmarks around this industrial zone where the subsidence rates from leveling are only 10 to 20 mm/a. However,

the InSAR result (Figure 14) shows large, evenly distributed subsidence rates in this industrial zone, with a maximum rate of about 82.3 mm/a.

*5.3. Area of Significant Subsidence and Rainfall: The Potential of Hushan Reservoir in Reducing Land Subsidence*

Here, our final discussion focuses on whether the InSAR technique has the potential to identify areas with significant subsidence and the relationship between land subsidence and rainfall. In Taiwan, the area of significant land subsidence (subsidence rate > 30 mm/a) is an indicator to judge whether a land subsidence prevention policy is successful. In order to reduce the use of groundwater and to mitigate land subsidence in Yunlin, the Taiwan government encourages farmers to use surface water supplied through waterways, and to grow crops less dependent on water [32]. In central Taiwan, the storage capacities of reservoirs have been substantially reduced by siltation in the reservoirs, resulting in decreased surface water supplies. To compensate for the reduction of surface water supply and shortages in the dry season, a new reservoir in Yunlin, called Hushan Reservoir, was opened in July 2016, and started to supply water in January 2019. Hushan Reservoir cost 0.66 billion USD to construct and its storage capacity is about 51,390,000 $m^2$ [33].

First, we examined the areas of significant subsidence in recent years using leveling data in order to see whether the government's past policies could mitigate land subsidence. Figure 16a shows the areas of significant land subsidence in Yunlin from 2011 to 2017 from leveling data [3] and annual rainfall and rainfall in the dry season from rain gauges [29]. As the leveling data were collected every April (once a year), an annual subsiding area represents the area of subsidence during the one year from the previous April to the current April. The rainfall in the dry season is the average rainfall from the previous October to the current April. Figure 16a shows that since 2013, the subsiding area increased and reached a maximum in 2015 (658.6 $km^2$). Note that 2015 was the record drought year in Taiwan. The rainfall in the dry season of 2015 (October 2014 to April 2015) was only 105 mm, compared to 158 mm in the dry season of 2014. Figure 16b shows the Oceanic Niño Index (ONI) from April 2010 to April 2017 [34], indicating that 2015–2016 is an El Niño year (with ONI above 0.5 in the five consecutive overlapping three months) that may be associated with the droughts and floods in 2015 in the Pacific coastal regions [35], including Taiwan (low rainfall in the dry season, Figure 16a). It is likely that the continuous drought in 2015 not only increased the magnitude of subsidence, but also widened the area of significant subsidence.

The rainfalls in Figure 16a suggest that, from 2011 to 2017, the areas of significant subsidence were negatively correlated with the dry-season rainfall (the correlation coefficient was –0.85). For example, in 2011, 2014, and 2017, the dry-season rainfalls were small, causing relatively large areas of significant subsidence (average: 360 $km^2$). By comparison, in 2012, 2013, and 2016, the dry-season rainfalls were larger and the areas of significant land subsidence were smaller (average: 122 $km^2$).

Both InSAR and leveling detected almost the same areas of significant subsidence in 2017 (Figure 16a; the difference was about 18 $km^2$). Table 3 shows the areas with subsidence rates >30 mm/a in Yunlin. Table 3 shows the area of significant subsidence from InSAR (215.5625 $km^2$) was close to the area from leveling (221.3325 $km^2$). The two differ by only 2.6%, suggesting that InSAR is reliable for detecting the area of subsidence. Figure 17 compares the areas of significant subsidence from InSAR and from leveling. Both results showed that Tuku, Yuanchang, and their nearby townships, experienced significant subsidence. The InSAR velocity map showed significant subsidence near Western Mailiao areas that were not seen in the leveling velocity map (see also Section 5.2). However, InSAR did not detect the full spatial extent of significant land subsidence in Southwestern Yunlin near Sihu Township, because this area is largely covered by rice fields (Figure 17a vs. Figure 17b).

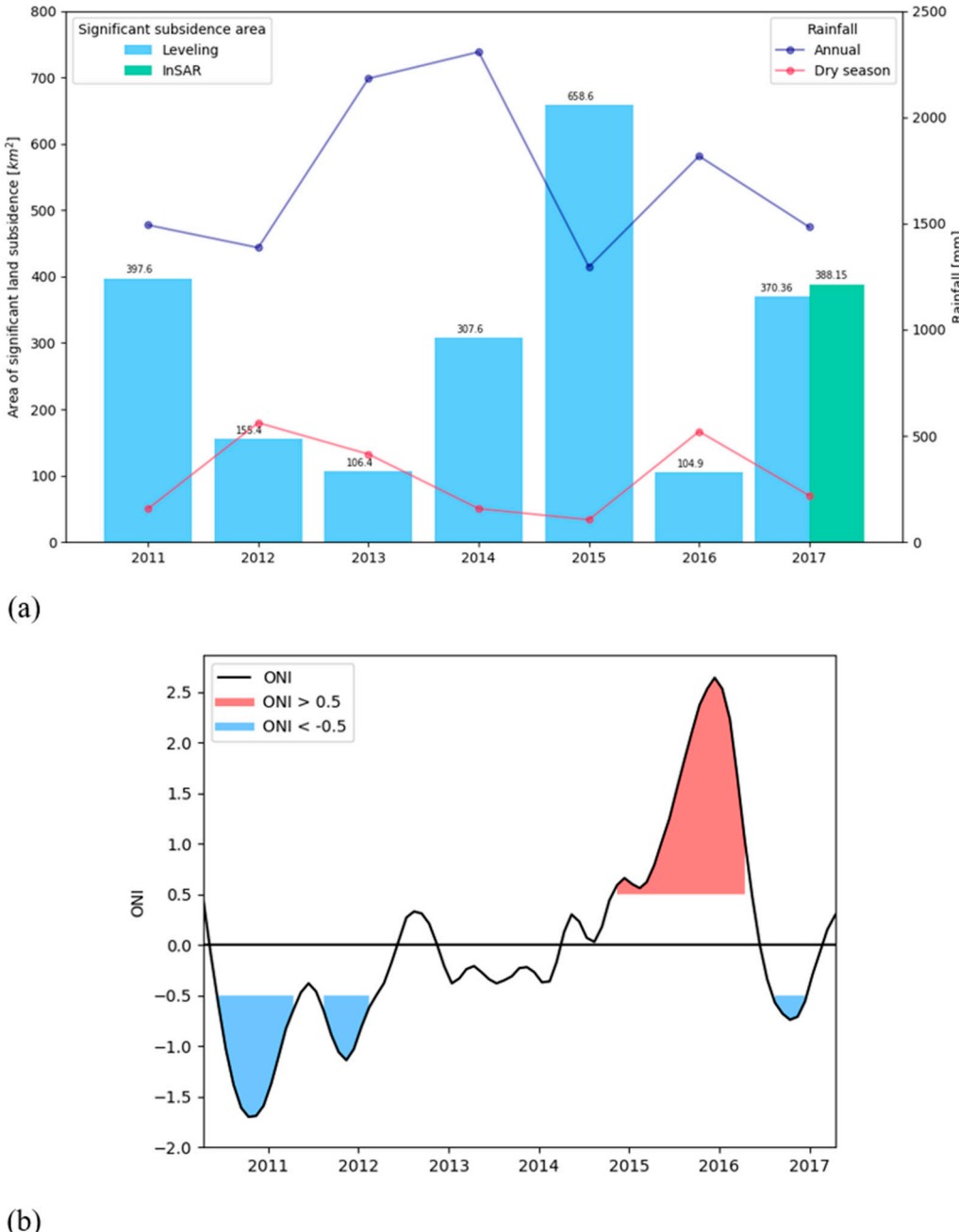

**Figure 16.** (**a**) Areas of significant subsidence from leveling (blue bars) from 2011 to 2017 [3] and from InSAR (green bar) in 2017 (this study), overlapped with annual rainfalls for the whole year (blue line) and the dry season (red line) in Yunlin [29]. (**b**) The ONI from April 2010 to April 2017 [34], the red-shaded period corresponds to the 2015 El Niño event and the blue-shaded periods show periods with ONI below −0.5.

As stated in Section 1, recent climate change has led to irregularities in the frequency, intensity, and magnitude of rainfall [2]. Such irregularities across seasons and across years (Figure 16) make it difficult to predict the amount of available surface water and short-term trends of land subsidence. Nevertheless, it is clear from the seasonal analysis in Section 5.1 that the subsidence in the dry season contributed the most to the annual subsidence (i.e., total subsidence in a year). Since Hushan Reservoir can supply surface water to counter the shortage of rain during dry seasons and droughts, it can potentially reduce the unpredictability of land subsidence in Yunlin.

**Table 3.** Areas with subsidence rates >30 mm/a from leveling and InSAR in 2017.

| Subsidence Rates (mm/a) | Leveling (km²) | InSAR (km²) |
|---|---|---|
| 30–40 | 221.3325 | 215.5625 |
| 40–50 | 100.9125 | 98.0775 |
| 50–60 | 37.9300 | 61.325 |
| 60–70 | 9.2275 | 11.9975 |
| over 70 | 0.9575 | 1.1875 |
| Significant subsidence area | 370.36 | 388.15 |

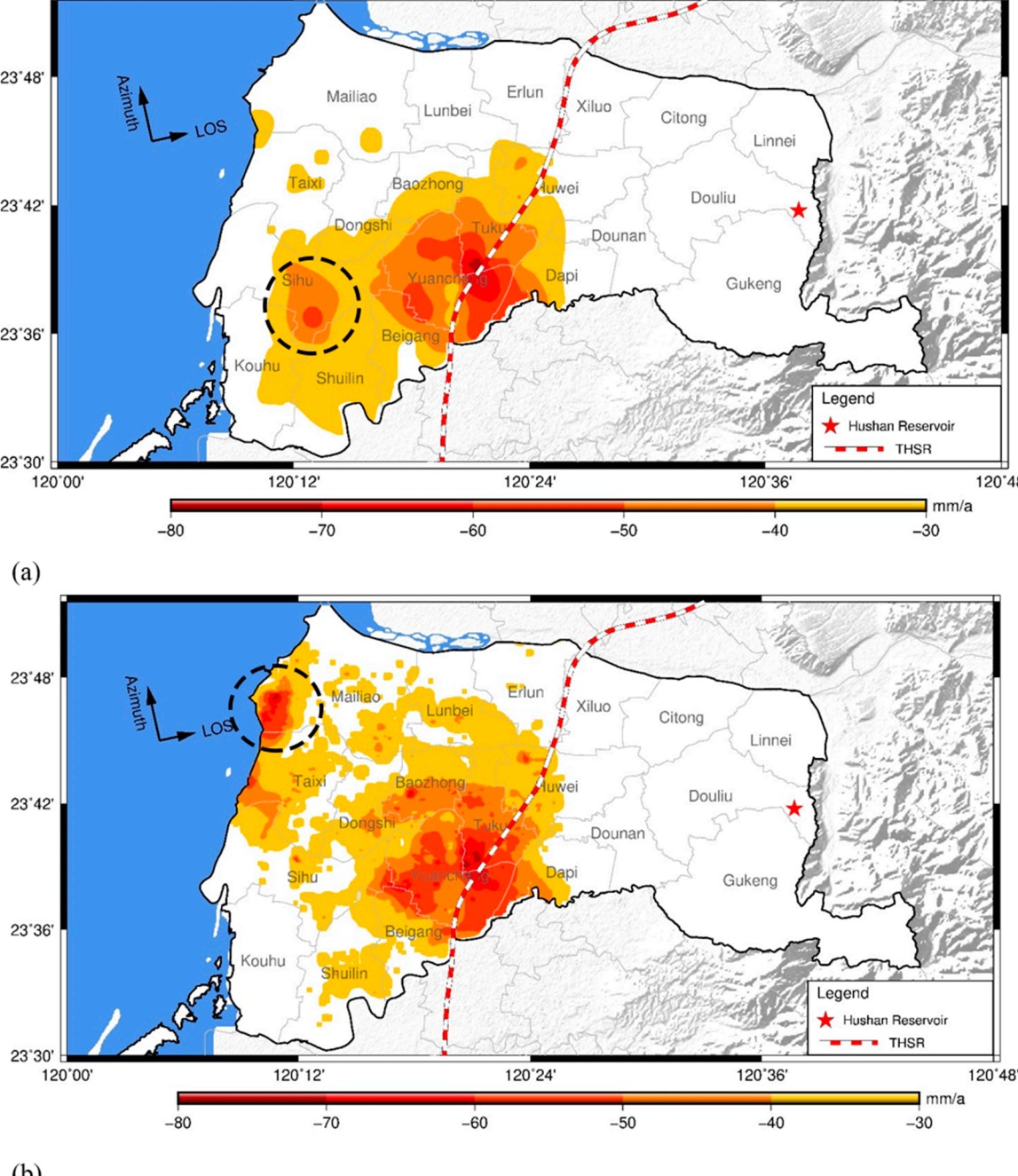

(a)

(b)

**Figure 17.** Areas of significant subsidence (rate >30 mm/a) from (**a**) leveling and (**b**) InSAR. The dashed circles show significant subsidence in (**a**) Sihu Township (strong signal from leveling, but weak signal from InSAR) and (**b**) Mailiao Township (detected by InSAR, but not by leveling).

## 6. Conclusions

This study shows the land subsidence monitoring results from April 2016 to April 2017 from Sentinel-1A InSAR, compared with GPS and leveling. The initial InSAR quality was improved by calibration using GPS measurements. Our monthly InSAR results indicated that land subsidence in the dry season contributed 60%–74% of the annual subsidence. This implies that more surface water should be supplied in the dry season than in the wet season to reduce groundwater pumping and land subsidence. Furthermore, InSAR identified two spots of severe subsidence not detected by leveling, highlighting the advantage of InSAR in showing detailed displacement information and in guiding government agencies to impose regulations to slow down newly developed subsidence.

Both InSAR and leveling identified almost the same areal extent of significant subsidence (rate > 30 mm/a) from April 2016 to April 2017 in Yunlin (the two differ by only 2.6% and the resulting rates were consistent at 10 mm/a). This suggests that InSAR is a reliable tool for detecting the occurrence of significant subsidence, which is an indicator of the effectiveness of a subsidence-mitigation measure. The variations in the area of significant subsidence in the past seven years indicate that rainfall in dry seasons is the deciding factor for severe subsidence. Our analysis suggests that the newly opened Hushan Reservoir may potentially reduce land subsidence by supplying surface water to counter the shortage of rain and to reduce groundwater pumping during dry seasons and droughts.

**Author Contributions:** Y.-J.Y. and C.H. initiated the idea in this study and wrote the manuscript, W.-C.H., T.F., Y.-A.C. and S.-H.W. provided data analyses and contributed with comments and suggestions to the writing of the manuscript.

**Funding:** This study is supported by Ministry of Science and Technology, Taiwan, under Grants 106-2221-E-009-133-MY3, 107-2611-M-009-001 and Water Resource Agency, Department of Economics, Taiwan, under Grant MOEAWRA1060345.

**Acknowledgments:** We thank ESA for providing the Sentinel-1A SAR images and David Sandwell and additional developers for maintaining the tool GMTSAR. This paper is published with the permission of the CEO, Geoscience Australia.

**Conflicts of Interest:** The authors declare no conflict of interest.

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
