# Peer review of "Surface Deformation from Sentinel-1A InSAR: Relation to Seasonal Groundwater Extraction and Rainfall in Central Taiwan"

_remotesensing, doi:10.3390/rs11232817_

Round 1
Reviewer 1 Report
Dear Authors, I have re-reviewed your manuscript, and I found numerous improvements. You have seriously considered all observations and accordingly modified the text. I think the paper is now suitable for publication.Author Response
We thank you for your nice comment. The paper is now further improved by taking into account the comments of Reviewers 2-4.
Reviewer 2 Report
Dear Authors,
in my opinion your paper is really improved.
Please, see the attached file for minor revisions.

Author Response
Thank you for your comments. Our responses (in red) are in the file rev2.pdf.

Reviewer 3 Report
Overall, the manuscript that employed the SBAS-InSAR technique for monitoring land subsidence in central Taiwan compared with GPS and in-situ measurement methods like leveling is well-structured, well-written and worth exploring. However, I have some observations regarding the tense used throughout the paper, and lack of references as significant InSAR land subsidence case studies were analyzed world-wide.
In the introduction, the authors clearly addressed the purpose of the study. However, on page 2- lines 45-55, all of the sentences require relevant references where authors explain the advantages and disadvantages of different monitoring approaches. It would also be great if authors can address in brief what are the problems/risks are associated with land subsidence, and follow on the importance/demand of this study in Yunlin. I appreciate the authors to include a summary section at the end of the introduction, and it helps readers to help them understand the structure of the manuscript. However, I suggest to re-write the section and change the future tense to present tense.
The method is well considered by selecting an advanced InSAR method based on multi-interferograms such as SBAS-InSAR as the traditional InSAR method has its limitation due to temporal decorrelation and atmospheric substance. In addition, it compares with other in-situ deformation monitoring techniques such as GPS and leveling, which is a clear indication for searching the best study results.
The results are clearly presented, and discussions are supported by the results. However, on page 20- in the last paragraph, you mentioned that the Hushan Reservoir can supply the surface water to counter the shortage of rain. Is there any study about this reservoir? If yes, please cite them to understand the capacity of the reservoir.
It would be helpful in the conclusion if the authors can add a few sentences mentioning the potential applications of land subsidence in formation in Yunlin, specifically with respect to sustainable spatial planning.
Specific comments:
Line 210: suggest April 2016- April 2017, to avoid confusion as it is used throughout the paper
Line 346: you use present tense throughout the discussion. Please be consistent
Line 349: suggest “remote sensing and subsurface data can help…..”
Line 407: please correct- “We examine the areas…..”
Line 430: suggest rewrite/change- “suggesting that InSAR is reliable…” instead of highly reliable
Author Response
Thank you for the comments. Our responses are in the file rev3.pdf.

Reviewer 4 Report
(please see the attached file)

Author Response
Thank you for your comments. Our responses are in the file rev4.pdf.

Round 2
Reviewer 2 Report
I have no further suggestions.
Best regards
Reviewer 3 Report
I would like to appreciate the authors' effort to revise the manuscript and improved it significantly. All comments addressed by the authors and the present format warrants publication in this journal. This is a good piece of a research article. No further comments- I appreciate the opportunity to review this manuscript.
Reviewer 4 Report
The authors have made the suggested revisions. I have no further insight to offer, as the manuscript and work presented within are of high quality.
This manuscript is a resubmission of an earlier submission. The following is a list of the peer review reports and author responses from that submission.
Round 1
Reviewer 1 Report
The manuscript by Yang et al. analyzes the surface deformation in Yunlin county (central Taiwan) using a small set of Sentinel-1A data and SBAS approach. The case history is significant because subsidence in the study region has strong implications for infrastructure management. InSAR data are calibrated using GPS stations and later compare to leveling measurements.
The Authors analyzed the April 2016 – April 2017 time interval and found that subsidence rates are controlled by the amount of rainfall during the dry season.
The approach is well-established and the main outcome is the identification of seasonal trends. My main concern regards the completeness of the dataset: 22 Sentinel 1A images (ascending track only) over a 1-year period are used. When integrating the leveling measurements, the analyzed time interval broadens to 2011-2017. I think this is not enough to detect secular variations, as stated in the title. Also the “previously unidentified” part of the title is not adequately addressed.
Some of the results/conclusions (e.g., lines 322-327; 353-358) are not supported by the data presented in the manuscript. I believe the results on the seasonal influence can be trusted, but a more statistically sound analysis of the data shown in figure 16 is needed.
Please note some format issues with the figures: Figs 3, 5 and 12 are repeated multiple times in the manuscript.
Overall, I think that the present paper is suitable for publication in a local journal; with some improvements, it may fit the standards of Remote Sensing journal.
Some additional comments are in the attached file.

Author Response
Please see the file response1.pdf.

Reviewer 2 Report
The manuscript titled “Detecting secular, seasonal and previously unidentified surface deformation by Sentinel-1A InSAR: dry-season rainfall is the deciding factor for land subsidence in central Taiwan” presents the approach to understand the subsidence in the wet and dry season periods. Overall, the manuscript needs to organized and writing can be improved. First, the importance of the manuscript is to understand the subsidence change in dry-wet season but there is zero or minimum background information on the sub-surface geology and hydro-geology. Second, if the author implies that rainfall is a factor for change in rate of subsidence at Yunlin, then why there is different spatial pattern in subsidence (Figure 17). Moreover, there is huge variations in the LOS displacement between wet and dry season (Fig 10), because within short period (one year) such variations are not convincing but the effect of poor processing and analysis. Therefore, I suggests the authors to revise the manuscript deeply and verify the SBAS time-series results before proceeding further.
Major comments:
Line 185: Sentinel-1 provides both ascending and descending data, which is very useful to resolve LOS displacements into two dimensional displacements For example, [1]. Because most of the GPS station shows considerable horizontal velocities (5 mm/a). Therefore, it is necessary to provide some information on InSAR derived horizontal velocities. Figure 12: The comparison results between GPS and InSAR velocities shows huge discrepencies, which suggests to check the reliability of InSAR results (Figure 12). If you think, this could be tropospheric delay, have you tried to remove these phase delays using methods such as GACOS, etc.,. Are you sure the interferograms are correctly unwrapped. Can you provide unwrapped phases? Line 151: On what basis, the author resampled the final LOS displacements into 297.5 m x 8.5 m? It is well-known that high resampling grid will average the noise of InSAR results, however this will also suppress the important information such as local deformation. This will further questions the reliability of the InSAR velocities. Figure 10: Please explain why there is significant uplift signal in Fig. 10a?
Minor comments:
Line18: Include sentinel-1A acquisition period Line 123: Please show the location of LNJS in Figure 1 Figure 3: The author have to revise the Figure 3, showing GPS data as point data not as line. Please check the Figure captions Also, provide period for figure showing LOS displacements.
References
Samieie-Esfahany, S.; Hanssen, R.; van Thienen-Visser, K.; Muntendam-Bos, A. In On the effect of horizontal deformation on InSAR subsidence estimates, Proceedings of the Fringe 2009 Workshop, Frascati, Italy, 2009.
Author Response
Please see response2.pdf.

Reviewer 3 Report
Dear Authors,
your paper is quite interesting, but, in my opinion it could be improved.
Geological and hydrogeological data of the study area are needed to better explain your statements on the causes of subsidence. You should consider to process the entire Sentinel-1 dataset available from 2014, both ascending and descending tracks. The results (velocity maps) of the processing are not fully convincing as well as the estimation of the vertical displacements (see also the comments in the text).Please, also consider the comments in the attached file.
Best regards

Author Response
Please see response3.pdf.
